# The media risk of infodemic in public health emergencies: Consequences and mitigation approaches

**Rui Shi[1], Xiaoran Jia[1], Yuhan Hu[1], Hao Wang [2]***

**1** School of Economics and Management, Yanshan University, Qinhuangdao, Hebei, China, **2** School of Economics and Management, Tianjin University of Science and Technology, Tianjin, China

\* wanghao47153434319@126.com

**Data Availability Statement:** All relevant data are within the manuscript and its Supporting information files.

**Funding:** This research was funded by the Science Research Project of Hebei Education Department

## Abstract

This study explores the nuances of information sharing in the context of infodemics, with a concentrated examination of the effects of opinion leaders and information attention on users' disposition towards sharing information during public health emergencies. The research adopts a quantitative methodology, employing Structural Equation Modeling (SEM) to empirically test the proposed hypotheses. By employing a rigorous analytical framework, the research also scrutinizes the mediating role of risk perception in shaping users' intentions to disseminate information related to public health emergencies. Additionally, it investigates the moderating effect of perceived usefulness, shedding light on how it influences the strength of the relationship between information attention and risk perception. The findings underscore the significance for public health communication strategies, emphasizing targeted messaging utilizing trusted opinion leaders and emphasizing information utility to foster responsible sharing. This research contributes to the academic conversation on infodemic management, providing empirical insights to guide policies and practices in mitigating misinformation during public health emergencies.

## Introduction

The intricate design and functionality of contemporary societies have become increasingly complex due to the synergistic effects of swift technological progress and economic globalization, rendering them more challenging to manage. Consequently, the frequency of public health emergencies has escalated, exemplified by major infectious disease outbreaks, epidemics of diseases with unidentified origins, and widespread food poisoning incidents [1–2]. These events are marked by their immediacy, unpredictability, systemic nature, and detrimental consequences [3], posing substantial challenges not only to the domains of medicine and science in terms of public health emergencies but also engendering unparalleled infodemic phenomena in the realm of information dissemination, creating an information crisis of historic proportions.

A significant challenge in mitigating and responding to large-scale risks is the phenomenon of the infodemic [4], a secondary risk associated with crisis management in media-saturated

[SQ2023105] in the form of a grant to RS. The funders had no role in study design, data collection and analysis, decision to publish, or preparation of the manuscript.

**Competing interests:** The authors have declared that no competing interests exist.

societies. At its core, it is a risk-mediating phenomenon that results from the contextual restructuring and evolution of emergency situations. Gunther Eysenbach initally proposed the term of infodemiology in 2002, aiming to investigate the distribution and determinants of health-related information, both accurate and erroneous, guiding patients and professionals in the field toward trustworthy and precise information [5]. Rothkopf defined the infodemic during the 2003 SARS outbreak as an event driven by contemporary information and communication technology and sparked by pandemics and other catastrophes that result in the rapid spread and proliferation of rumors, panic, and false information [6]. Since the start of the COVID-19, online social media platforms have allowed the infodemic, which is typified by false, misleading, and malicious information, to spread throughout the world [7]. These platforms serve as primary information channels that accelerate distribution and expand reach due to their decentralized nature; however, they also enable the proliferation and propagation of false information [8]. Algorithmic recommendations may inadvertently amplify the circulation of deceptive information [9], while patterns of user information sharing and tendencies toward group polarization further exacerbate the issue by reinforcing echo chambers and information bubbles [10].

In addition to denoting the excessive, rapid, and overwhelming spread of conflicting information in high-risk situations, the infodemic is closely associated with the public's changing psychology and behavior [11]. In the internet age, where new media are taking center stage, social media users exhibit a preference for gathering information on significant public health emergencies from diverse sources. However, their inability to ascertain the veracity of this information and the reliability of its originators leaves them vulnerable to developing convoluted perceptions of uncertainty. Consequently, societal hazards are exacerbated as these risk perceptions permeate through interpersonal networks and public psychological responses, becoming embedded in the collective consciousness. The subsequent psychological reactions and interactions of the public reinforce these perceptions, intensifying the threats to societal stability. The shaping of public risk perception by the diffusion of risk information is a critical factor in dictating human behavior and communication regarding risk in emergency situations [12]. Risks are typically not encountered directly; instead, they are perceived through a "virtual environment" created by the news media. Kasperso et al. illustrate a concrete pathway by which risk signals are amplified via social magnification mechanisms, notably through social media platforms, and influential opinion leaders functioning as individual amplifiers, either bolstering or attenuating public awareness of risks [13]. Several studies have elucidated the function of media in magnifying risk perceptions at the individual level during this process [14–17]. Nevertheless, the outcomes vary contingent upon the distinct attributes of the communicators and the nature of the information disseminated.

The propagation of information within infodemics is primarily propelled by public information processing and sharing behaviors, which significantly influence the trajectory of the epidemic [18]. From a communication standpoint, social media users within the framework of social networks represent diverse, dual roles: they are simultaneously information consumers and recipients, as well as information disseminators and sharers [19]. Acting as information seekers, users proactively scour and sift through data from a multitude of sources, and in their capacity as disseminators, they relay content deemed significant to fellow members of their social networks [20]. Information screening, appraisal, and sharing are markedly shaped by psychological elements (e.g., fear, anxiety, and trust), the extent of attention devoted to emergencies information, and the subjective evaluation of information [21]. Numerous social risks escalate into acute conflicts due to discrepancies in risk perceptions among different stakeholders, leading to divergent coping strategies and behaviors that stem from these perceptual disparities.

In summary, the intricate dynamics between social media users' information sharing behaviors amidst infodemics and their perceptions of public health emergencies risks forms a complex system entailing individual psychology, social interactions, and the information ecosystem. Set against this backdrop of risk information dissemination during infodemics, opinion leaders within social media landscapes, exploring their mechanisms of influence on audience cognition and information sharing willingness. Moreover, it delves into the manner in which individual information processing styles influence how behavioral intentions and risk perception are related. By illuminating users' information handling approaches, the study aims to predict behavioral decision responses and inclinations, thereby informing the formulation of emergency information management strategies by government bodies and proposing recommendations to address online public opinion crises during public health emergencies. This undertaking is directed at addressing societal concerns and mitigating public anxiety.

## Hypotheses development

### Opinion leaders and information sharing willingness

Against the backdrop of public health emergencies, such as the COVID-19 and the H1N1 influenza outbreak, social media has emerged as a pivotal platform for the public to access information, share personal experiences, and express emotions. Opinion leaders, with substantial followings and influence over public sentiment, wield their expertise, extensive information networks, and active digital engagement to bolster the authority and persuasive power of their discourse. This significantly shapes public thought and action, highlighting their substantial sway [22]. The urgency and unpredictability that accompany unexpected public health emergencies heighten public curiosity while also escalating the threat of disinformation and rumors spreading [23]. Opinion leaders' involvement adds another layer of complexity to this dynamic [24]. Their positive recommendations have the power to strengthen the social proof effect of content and increase users' propensity to spread [25]. Conversely, unfavorable assessments or skepticism expressed by these leaders may restrain sharing activities, particularly within an environment clouded by informational ambiguity [26]. Synthesizing these observations, grounded in the pivotal function of opinion leaders, the following research hypothesis is posited:

H1: Opinion leaders, leveraging their influence, specialized knowledge, and credibility, positively affect on users' willingness to share information.

### Information attention and information sharing willingness

Information attention directly affects the depth of information processing and the development of cognitive frameworks. It is defined as the extent to which users constantly track and delve into information surrounding a particular emergency occurrence [27]. Xue categorizes user information behaviors into two primary dimensions: information sharing and information attention [28]. In an environment saturated with information, individuals prioritize selective attention before sharing, progressively sharing content that is both significant and engaging to peers in an inclusive relationship that fosters sharing as a result of sustained attention. During an infodemic, individuals who focus on the same event develop an emotional and cognitive bond, motivating them to exchange knowledge in support of collective interests and the dissemination of consensus [29]. Increased awareness of information allows for deeper analytical processing, which includes fact-checking and critical analysis [30]. In addition, the deep understanding gained through this focused attention enables individuals to discern their role and function within the continuum of information dissemination, instilling a sense of

responsibility as "accountable information citizens." This heightened awareness promotes a more cautious approach to the selection and sharing of information [31]. Drawing upon the theory of cognitive investment, which posits that intensive processing and sustained attention enhance the perceived value of information and subsequently stimulate sharing intentions, we propose the following hypothesis:

H2: Users' attention to information exerts a positive influence on their information sharing willingness.

## Mediating role of risk perception in the link between opinion leaders and information sharing willingness

Research continuously confirms that an individual's perception of risk significantly impacts their evaluation and reaction to crisis situations [32]. According to the psychometric paradigm proposed by Slovic and colleagues, risk perception—a subjective interpretation and assessment of risk attributes and event severity—is shaped by both emotional and cognitive factors [33,34]. In the absence of direct knowledge about emergencies, the public often turns to social media platforms for real-time insights and updates [35]. Opinion leaders, who possess specialized knowledge, industry experience, or societal influence, are frequently seen as trustworthy and authoritative sources. Their statements and shares carry significant weight during crises, as people seek reliable information to guide their actions, making the messages from opinion leaders critical in risk assessment [30]. Following the dissemination of risk-related information by opinion leaders, the general public forms attitudes and opinions based on their own interpretations and understanding of the circumstances. If the public perceives the risk as severe or directly relevant to them, they may feel compelled to share the information widely as a collective response to mitigate the threat [36]. Accordingly, the following hypothesis is postulated:

H3: Risk perception acts as a mediator in the relationship between opinion leaders and information sharing willingness, and it positively impacts users' willingness to share information.

## Mediating role of risk perception in the link between information attention and information sharing willingness

The disorderly and chaotic dissemination of information frequently dilutes valuable content amidst a sea of trivial data. This not only escalates the risks associated with information reception, but also amplifies the likelihood of information hazards due to the proliferation of inferior, distorted, and ineffective messages [37]. Social media users are more prone to interact with risk-related content when faced with perilous situations, as heightened subjectivity and complex information processing mechanisms are activated. Taking the COVID-19 as an illustration, the pathogen's unfamiliarity, combined with its extreme contagiousness and compounded by information overload and distortion, renders subjective factors decisive in shaping risk perception. Attention to the event significantly impacts cognition and psychological representations [38]. Depending on individual needs, people perceive a variety of risk attributes differently, such as familiarity, controllability, and knowledge, which influences their judgment of the severity of a risk [39]. Increased awareness of risk motivates people to actively search for, focus on, and carefully consider pertinent information to mitigate uncertainty and create self-protection strategies. This increased attention to information, mediated by risk perception, increases the inclination to communicate information, facilitating alert issuance,

preventive measure distribution, or seeking assistance from others [40]. Hence, the following hypothesis is posited:

H4: Risk perception mediates the association between information attention and information sharing willingness, positively affecting users' willingness to share information.

## Moderating role of perceived usefulness between information attention and risk perception

The Technology Acceptance Model (TAM), introduced by Davis in 1989 [41], represents an adaptation of the Theory of Reasoned Action specifically designed for the Information Systems domain. According to this paradigm, perceived utility and perceived ease of use are posited as critical determinants of users' behavioral intentions. In the context of information processing, users employ a series of filters to evaluate and ultimately accept information that they deem relevant to their decision-making processes, based on its perceived utility. Information with high perceived utility accelerates and enhances the accuracy of decision-making [42]. Notably, under conditions of information overload, particularly in emergency scenarios, individuals prioritize attending to and processing information they perceive as beneficial to their personal safety and survival strategies [43]. Such high-utility information is more captivating, stimulating meticulous processing and influencing risk assessments. Conversely, when information is perceived as lacking utility or relevance, it can lead to distorted risk perceptions, resulting in either an overestimation or underestimation of threats. Accordingly, the following hypothesis is postulated:

H5: Perceived usefulness of information exerts a moderating effect on the relationship between information attention and risk perception.

## Research method and design

### Structural equation modelling

Structural Equation Modeling (SEM) stands as a highly versatile and comprehensive statistical analysis technique and framework, supplanting methods such as multiple regression, path analysis, factor analysis, and covariance analysis. It enables a lucid examination of both the individual impacts of indicators on the overall construct and the interrelationships among these indicators. SEM not only manages causal interactions among observable variables but also facilitates the construction of multiple "latent variables" through factorial analysis, thereby delving into intricate relationships between latent variables or between latent and observed variables [44].

In the application of SEM for analytical purposes, it is imperative to presuppose the directional paths and connections within the conceptual model's framework. Based on literature review, the conceptual model diagram is shown in Fig 1. Among them, opinion leaders and information attention as dependent variables, information sharing willingness as an independent variable, risk perception as a mediating variable, and perceived usefulness as a moderating variable.

Prior to conducting our study, we received approval from the Ethics Committee of Qinhuangdao First Hospital, Hebei Province, China. Participants signed an informed consent before the experiment. The data were collected anonymously and analyzed in aggregated form.

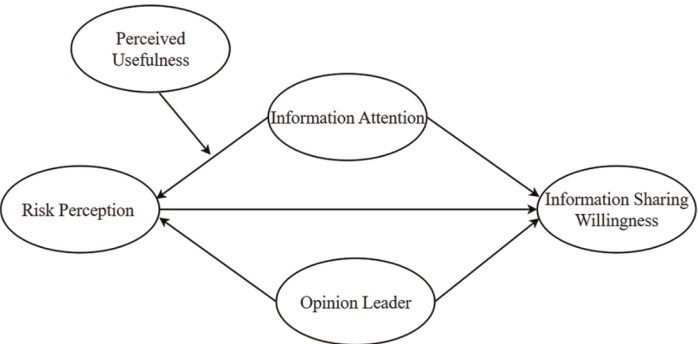

**Fig 1. The structural model.**

## Survey instrument

The questionnaire includes five latent variables: information attention, opinion leaders, risk perception, information sharing willingness and perceived usefulness. Among them, information attention (IA) questions refer to the Xu(2020) [45] and Powell D(1996) [46]; opinion leaders (OL) questions refer to the Wang(2018) [47]; risk perception (RP) questions refer to Slovic P(1992) [48]; information sharing willingness (ISW) questions refer to Bock GW et.al(2012) [49]; perceived usefulness (PU) questions refer to Maltz(2000) [50]. In concordance with the study hypotheses and mindful of the exigencies dictated by the COVID-19, a meticulous revision and contextualization of survey items sourced from antecedent investigations was undertaken. Precisely, within the rubric measuring information attention, pandemic-specific nuances were integrated, exemplified by the item reflecting vigilance over the evolving dynamics pertinent to the COVID-19. Concerning risk perception, the awareness of potential perils intrinsic to the COVID-19 was emphasized through the calibration of relevant questions. Furthermore, in the evaluation of proclivity towards information sharing willingness, the pertinence of shared information to the COVID-19 milieu was explicitly delineated, as embodied in the item highlighting the propensity to disseminate or alert others upon receipt of updated the COVID-19 information. Through these targeted refinements, the questionnaire's contemporaneity and situational specificity were assured, thereby facilitating a more accurate delineation of participant behavioral paradigms and psychological orientations situated within the COVID-19 framework. Specific scales for each of these variables can be found in the support information, please see the S1 Appendix section for details.

Each variable is designed with 3–5 questions, totaling 18 items. A five-level Likert scale is a widely used research tool in sociology and management. The scale usually contains five rating options, each representing a different intensity of attitude. It is a common tool used to measure a participant's attitude towards a particular issue. The variables in this study were all measured using a five-level Likert scale method, using integers between 1 and 5 to measure the participants' attitudes towards the questions. The rating of 1 is strongly disagreeing, and 5 is strongly agreeing. Participants choose based on their actual situation.

To ensure the accuracy of the data, a pre-survey was conducted before the official release of the questionnaire. 110 questionnaires were distributed to the pre-survey purchase and 104 valid responses following the exclusion of 6 invalid ones. Subsequent analysis affirmed both the reliability and validity of the questionnaire, alongside acceptable model fit within the structural equation modeling framework. This substantiates the rationality and authenticity of the collected data.

## Sample and data collection

Questionnaires were disseminated online from March 1 to March 7, 2024, via WeChat and the Credamo platform. This digital outreach was designed to maximize accessibility and inclusivity, thereby engaging a diverse participant base.

To ensure the representativeness of the research sample, this study adopted an integrated sampling strategy that artfully combined snowball sampling with random sampling methods. This approach aimed to construct a sample framework that accurately reflects the structure of internet users, particularly in terms of gender, age, and educational background. The detailed methodology is outlined as follows: surveys were distributed via stratified random sampling among various age and gender groups in the researchers' social networks, asking participants to haphazardly pass the questionnaire to their friends to fill out. Data collection from demographic categories that were properly represented was then stopped once a preset amount of responses were received. Lastly, to counteract the bias of snowball sampling, targeted random distribution of questionnaires was conducted on the Credamo platform for underrepresented groups. Respondent selection enhanced sample diversity, while accompanying detailed questionnaire explanations secured participant understanding, boosting response quality and data validity, and solidifying research reliability. This multi-faceted approach was instrumental in compiling a well-balanced and informative dataset for the study.

For exploring variable relationships via structural equation modeling, a minimum sample size of 200 is advisable, ideally exceeding ten times the number of observed variables for robust analyses [51]. In this study, 395 questionnaires in total were sent; 346 of those were deemed legitimate, and the effective recovery rate was 87.59%.

Of the 346 participants, 45.09% were men and 54.91% were women, indicating a relatively balanced gender ratio. In terms of age, the participants were all over 18 years of age, most of them were in the 18–45 age range. Among the participants, 10.11% had only a high school diploma or less, 38.73% had associate degrees, 42.20% held bachelor's degrees, 7.51% were pursuing or had completed master's degrees, and the remaining 1.45% were doctoral degree holders. The demographic characteristics of the sample are shown in Table 1. The examination of the comprehensive distribution of our surveyed sample is consistent with the profile of social media platform users outlined in Quest Mobile's *2023 Annual Report on Core Trends in China's Internet* [52], reinforcing the representativeness and alignment of our sample with current internet demographics.

Table 1. Demographic characteristics of the sample.

| Demographic Variables | Categories | Frequency (n = 346) | Percentage (%) |
|---|---|---|---|
| Gender | Male | 156 | 45.09% |
| | Famle | 190 | 54.91% |
| Age | 18–25 | 144 | 41.62% |
| | 26–35 | 107 | 30.92% |
| | 36–45 | 49 | 14.16% |
| | 46–55 | 25 | 7.23% |
| | 56–65 | 16 | 4.62% |
| | Over 66 | 5 | 1.45% |
| Education Level | High school or less | 35 | 10.11% |
| | Associate degree | 134 | 38.73% |
| | Bachelor' s degree | 146 | 42.20% |
| | Master' s degree | 26 | 7.51% |
| | Doctor degree | 5 | 1.45% |

## Data analysis

The minimum anonymous dataset required for replicating the study results can be found in the support information, please see the S1 Dataset section for details.

## Results of reliability and validity

Due to measurement errors, the reliability and validity of the questionnaire are tested before the data analysis, ensuring research outcome credibility. Among them, reliability refers to the degree of reliability of the measurement data, and validity refers to the degree of agreement between the observed variables and the latent variables. In this study, SPSS 24.0 software was used to test the reliability and validity, and the data are shown in Table 2. The reliability of the scale was tested by using Cronbach's $\alpha$ coefficient, and the reliability coefficient of the overall questionnaire was 0.898. And each latent variable has reliability coefficients greater than 0.78, indicating that overall reliability of the scale is relatively reliable. For the validity test, the KMO value of the output data is 0.894, and the KMO values of the five latent variables are grater than 0.7, which indicates that the correlation between variables is strong and suitable for factor analysis. According to the standard value, all latent variables demonstrated composite reliabilities (CR) above 0.6, alongside average variance extracted (AVE) values greater than 0.5, indicative of sound internal consistency and satisfactory convergent validity. Furthermore, the square root of each AVE exceeded its corresponding correlation coefficients, as evidenced in Table 3, attesting to the discriminant validity of the research data. Collectively, these metrics attest to the high quality of the dataset, warranting further advanced analyses.

## Path analysis and hypothesis testing

This paper endeavors to employ the AMOS 24.0 software, a prominent tool in SEM analysis, to calibrate and refine a theoretical model explicating the determinants influencing the information sharing willingness on public health emergencies. The model fit indices are presented in Table 4, with the data indicating that the theoretical model exhibits a good fit, as all metrics

**Table 2. Reliability results.**

| Observed Variable | Standard Factor Loading | Cronbach's $\alpha$ | CR | AVE |
|---|---|---|---|---|
| OL1 | 0.800 | 0.810 | 0.813 | 0.594 |
| OL2 | 0.671 | | | |
| OL3 | 0.832 | | | |
| IA1 | 0.774 | 0.848 | 0.853 | 0.531 |
| IA2 | 0.773 | | | |
| IA3 | 0.730 | | | |
| IA4 | 0.725 | | | |
| IA5 | 0.656 | | | |
| RP1 | 0.716 | 0.819 | 0.809 | 0.537 |
| RP2 | 0.743 | | | |
| RP3 | 0.807 | | | |
| RP4 | 0.657 | | | |
| ISW1 | 0.781 | 0.788 | 0.773 | 0.533 |
| ISW2 | 0.743 | | | |
| ISW3 | 0.662 | | | |

**Table 3. Discriminant validity.**

|  | IA | OL | RP | ISW |
|---|---|---|---|---|
| IA | 0.771 | | | |
| OL | 0.338 | 0.729 | | |
| RP | 0.392 | 0.239 | 0.733 | |
| ISW | 0.505 | 0.268 | 0.238 | 0.730 |

fall within acceptable ranges [53]. Utilizing the Maximum Likelihood Estimation method, the path coefficients among variables within the model were estimated, and the hypothesis testing results are detailed in Table 5. As visualized in Fig 2, the final path coefficients are depicted, adhering to academic conventions where ellipses symbolize latent variables, circles denote residuals, and unidirectional single-headed arrows signify causal relationships. It is noteworthy that exogenous latent variables are not accompanied by residuals, whereas endogenous ones incorporate them. The empirical findings reveal that each path coefficient value resides between 0 and 1, and the corresponding t-tests yield statistically significant results at the 0.05 level, thereby confirming the establishment of hypotheses H1, H2, H3, and H4.

## Mediating effect and moderating effect

In order to verify the mediating role of risk perception in the model, this paper adds the risk perception variable to the information sharing willingness model to test whether it has a mediating role between users' information attention, opinion leaders and sharing willingness. On the basis of regression analysis, applying Process 3.5 plug-in and Bootstrap method [54], model 4 in the Process programme was selected, 5000 repetitions of sample extraction were performed, and the mediating effect was analysed with 95% as the significance confidence interval, and significance test was performed. The upper and lower bounds of the confidence interval are used in the mediation effect analysis of risk perception to assess the degree of influence between variables. A substantial mediation effect is found when there is no zero in either of the confidence interval's higher or lower bounds. Table 6. demonstrates that information attention and opinion leaders have the potential to influence a user's willingness to share

**Table 4. Model fit.**

| Indices | CMIN/DF | RMSEA | GFI | AGFI | CFI | NFI |
|---|---|---|---|---|---|---|
| Standard for adaption | <3 | <0.08 | >0.9 | >0.9 | >0.9 | >0.9 |
| Test result | 2.307 | 0.062 | 0.929 | 0.898 | 0.951 | 0.917 |

**Table 5. Hypotheses test results.**

| Path hypothesis | | | Standardized estimate | S.E | P | Test result |
|---|---|---|---|---|---|---|
| Risk perception | → | Information attention | 0.392 | 0.091 | *** | Significant impact |
| Risk perception | → | Opinion leaders | 0.239 | 0.061 | *** | Significant impact |
| Information sharing willingness | → | Risk perception | 0.238 | 0.051 | *** | Significant impact |
| Information sharing willingness | → | Information attention | 0.505 | 0.072 | *** | Significant impact |
| Information sharing willingness | → | Opinion leaders | 0.268 | 0.043 | *** | Significant impact |

(***P<0.001)

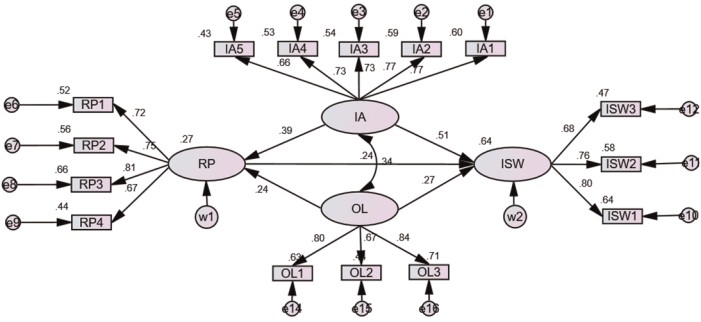

**Fig 2. Results of the path analysis.**

information not just directly but also through the mediating effect of risk perception. (Path 1: Opinion leaders → Risk perception → Information sharing willingness; Path 2: Information attention→Risk perception→Information sharing willingness).

To scrutinize whether perceived usefulness moderates the relationships between information attention and risk perception, following an initial analysis of the main effect pathways in the model, we adopted the hierarchical regression analysis approach recommended by Wen et al. to examine the moderating role of perceived usefulness [55]. Firstly, to mitigate multicollinearity issues, the independent variable, information attention, was centered. Next, in the first-step regression model, we incorporated the independent variables, the moderator, and the dependent variable. Proceeding to the second step, an interaction term, derived from multiplying the independent variable with the moderator, was included in the regression equation. The presence of a significant interaction term coefficient would signify the existence of moderation. Significantly, the interaction between information attention and perceived usefulness exerted a significantly influential effect on perceived risk (p = 0.0063<0.01), thereby affirming that perceived usefulness indeed moderates the interplay between information attention and its own construct. Consequently, this empirical evidence substantiates the validation of H5.

To visually illustrate the moderating effect of perceived usefulness on the relationships between variables, a simple slope analysis was conducted using the mean ± one standard deviation (M±SD) as the grouping criterion, depicting the interplay under differing risk communication contexts. The resultant illustration is presented in Fig 3, where the red and blue lines represent the varying strengths of the impact between variables at differing levels of perceived usefulness. Specifically, the slope of the red line exceeds that of the blue line, signifying a stronger positive influence of information attention on information sharing intention under conditions of high perceived usefulness. In essence, this finding underscores that the impact of information attention on risk perception is augmented as social media users' perceived

**Table 6. Mediatior analysis.**

| Path | Effect Type | Estimate | Boot SE | Boot LLCI | Boot ULCI | Relative effect |
|---|---|---|---|---|---|---|
| Path 1 | Total effect | 0.383 | 0.043 | 0.298 | 0.469 | |
| | Direct effect | 0.284 | 0.046 | 0.201 | 0.368 | 74.15% |
| | Indirect effect | 0.099 | 0.030 | 0.044 | 0.594 | 25.85% |
| Path 2 | Total effect | 0.670 | 0.050 | 0.573 | 0.768 | |
| | Direct effect | 0.555 | 0.051 | 0.456 | 0.654 | 82.84% |
| | Indirect effect | 0.115 | 0.031 | 0.042 | 0.182 | 17.16% |

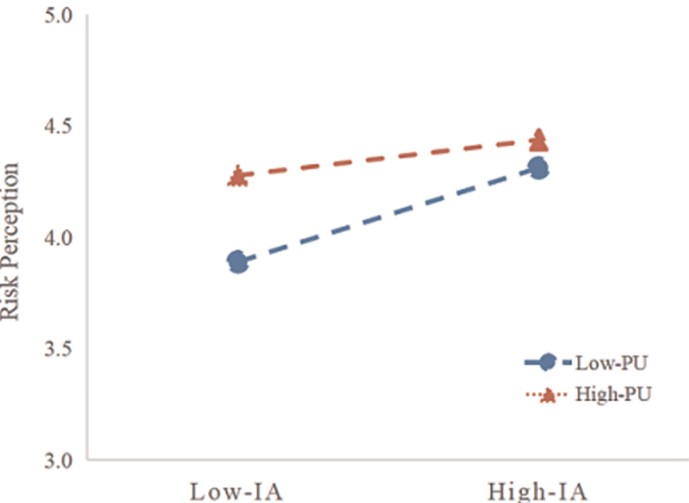

**Fig 3. Moderating effect.**

usefulness of the information they encounter increases, highlighting a synergistic effect between the information attention and its perceived value in shaping risk perceptions among users.

## Results and discussion

### Positive impact of social media on information sharing willingness

The impact of social media on users' information-sharing behaviors is multifaceted, encompassing dimensions of individual conduct, social interaction, information dissemination, and psychological mechanisms. The empirical results corroborate the significant influence of opinion leaders on social media platforms in shaping users' propensity to share information, in alignment with the findings previously reported by Zhao [56] and Bakshy [57]. Opinion leaders increase the trustworthiness and appeal of information by sharing high-quality and valuable content, which encourages people to share and participate in conversation. Moreover, the active engagement they have with their followers—shown by their quick likes, comments, and shares of content created by their followers—builds a sense of loyalty and trust. This emotional connection turns out to be a crucial information-dissemination trigger.

Furthermore, the results support the assertion by Xia [58] that heightened attention to information positively influences with an increased propensity to share. When users perceive that they have meticulously scrutinized and validated the accuracy and public benefit of a piece of information, their inclination to share it escalates, as sharing is now seen as an act of duty and contribution. It is noteworthy, however, that information overload emerges as a potential countervailing force against information attention [59]. Amidst the deluge of data characteristic of the digital era, excessive information can divert individual focus, thereby diminishing attention to critical information and potentially suppressing sharing activities due to cognitive overload.

Within the social media landscape, the actions of opinion leaders intertwine with users' levels of information attention, collaboratively shaping patterns and intensities of information dissemination. Leveraging their authority, emotional resonance, and agenda-setting capabilities, opinion leaders steer user focus towards specific information, while elevated attention, in

turn, amplifies the desire to share, fostering a feedback loop from attention to dissemination. The dual challenges of information overload and misinformation underscore the growing reliance on opinion leaders for content curation and validation, as well as the need for individual discernment in processing information prior to sharing decisions. In conclusion, the dynamics of information sharing within social media ecosystems emerge as a product of the interplay between the sway of opinion leaders and the level of attention accorded to information. This intricate relationship exerts profound implications for the efficiency, reach, and societal impact of information diffusion, offering a pivotal lens through which to comprehend the intricacies of the information ecology.

## The mediating role of risk perception

The results of the study show that risk perception acts as a mediator between opinion leaders and information attention and favorably affects people's willingness to share information about public health emergencies. In times of crisis, there is a sharp increase in the public's demand for risk-related information due to their pressing need to remain up to date on the most recent advancements, preventive measures, treatment options, and other relevant information in order to make decisions that will protect their own and their families' well-being [60]. People are more likely to actively seek out and share pertinent information when they perceive a larger level of risk, which satisfies their informational demands. Sharing helps people understand the problem better on a personal level, but it also creates a social network setting where information is shared, which makes society as a whole more resilient. Moreover, the public's assessment of the possibility of risk connected with emergencies is influenced by their trust in reputable organizations and professionals, especially opinion leaders, and this influences their decision to share information in accordance with that assessment [61]. The public views these leaders as reliable sources and pays closer attention to their remarks and advice when risk perceptions are elevated. This trust encourages a more proactive approach to information sharing, assisting others in comprehending the epidemic, increasing consciousness, and reducing the risk of transmission.

## The moderating role of perceived usefulness

The research reveals that perceived usefulness positively moderates the relationship between users' attentiveness to public health emergencies information and their risk perception. This implies that, prior to the formation of an intention to share information, it is crucial to establish a favorable perception of the information's utility. In an online environment saturated with information overload and misinformation, where trust in information is challenging to establish, neither the quality of the information nor the credibility of its source directly impacts user behavior; instead, these factors operate through perceived usefulness, first fostering cognitive acceptance, and subsequently influencing behavioral outcomes. Moving on to, the positive effect of information attentiveness on sharing willingness is strengthened under conditions of high perceived usefulness, whereas it weakens when perceived usefulness is low. A plausible underlying mechanism for this pattern could be attributed to the intricate context of infodemics, where social media platforms act as hubs for information aggregation, filled with a myriad of content. Users, guided by personal needs and evaluations of information utility, sift through this content to identify information they perceive as practically beneficial. Such information prompts a stronger cognitive alignment and deeper engagement from users, with this augmented sense of alignment and focus depth fueling an intrinsic motivation to share this category of information.

## Conclusions

The exploration of the mechanisms by which social media influences user risk perceptions and information sharing intentions under the context of infodemics carries profound theoretical and practical implications.

Theoretically, this research enriches our understanding of the complex dynamics between technology use, particularly social media platforms, and human behavior in times of crisis. By delving into the cognitive and psychological processes that underpin risk perception and information dissemination, it contributes to the advancement of communication theories, including infodemiology, risk communication, and social amplification of risk frameworks. It also elucidates the role of opinion leaders in shaping public discourse and highlights the significance of individual differences in information processing as critical moderators in risk communication, thereby refining our knowledge of information behavior in the digital age.

Practically, the study holds significant value for policymakers, public health officials, and crisis management teams. Insights gained can inform the development of more targeted and effective communication strategies aimed at mitigating the negative effects of infodemics. Understanding how social media shapes risk perceptions can guide the creation of tailored messaging that enhances public trust, combats misinformation, and promotes responsible information sharing. Moreover, by identifying the unique contributions of opinion leaders and the impact of user information processing styles, the research offers actionable guidance for leveraging influential voices in social networks to disseminate accurate and timely information, fostering a more resilient and informed society.

This research makes unique contributions by integrating multi-disciplinary perspectives to provide a comprehensive analysis of the infodemic landscape. It moves beyond descriptive analyses to offer a nuanced examination of the causal pathways linking social media exposure, individual cognition, and collective behavior. By highlighting the intricate interplay among social media environments, risk perception, and information sharing, it underscores the necessity for a comprehensive, bidirectional approach to risk communication that addresses not only the spread of misinformation but also fosters an ecosystem conducive to the accurate and responsible exchange of information. Ultimately, this work paves the way for evidence-based interventions that can effectively navigate the challenges posed by infodemics in future public health emergencies.

## Recommendations

Social media play an important role in the dissemination of information on public health emergencies. The implementation of appropriate measures to prevent and mitigate the escalation of public health emergencies is significantly enhanced by access to accurate and factual information. This enables individuals to develop a realistic understanding of the risks involved and gauge the seriousness of such emergencies accurately. However, the proliferation of false information can lead to heightened fear and panic, causing individuals to misinterpret the severity of the situation and adopt harmful behaviors based on misinformation. This not only undermines public health efforts but also takes a toll on mental health. By understanding the dynamics of information spread, public health authorities can develop targeted interventions. These strategies may involve:

**Enhance Transparency and Credibility.** Official sources should prioritize timely release of objective, verified information to counter misinformation and foster an environment where individuals can form accurate risk perceptions. This includes leveraging social media platforms for direct, transparent communication with the public.

**Risk Communication Strategies.** Develop and implement risk communication strategies that account for the complexity of infodemics, emphasizing clear, concise messaging tailored to different audiences and their information processing habits. This includes acknowledging and addressing fears and uncertainties associated with public health crises.

**Empower Opinion Leaders and Influencers.** Collaborate with credible influencers and community leaders on social media to amplify accurate information and promote positive behavioral change. Training programs can be initiated to enhance their understanding of public health emergencies and effective communication techniques.

**Psychometric Analysis.** Utilizing psychometrics enables the identification of vulnerable populations who may be more susceptible to misinformation and its associated stress effects. By understanding the psychological profiles of individuals prone to anxiety and misinformation acceptance, targeted interventions can be designed. These might include mental health support services, fact-checking resources, and community-led initiatives that foster critical thinking and resilience.

## Research limitations

While the present investigation has yielded valuable insights and practical applications regarding the determinants of information sharing intentions in the context of public health emergencies, it is not without its constraints. These limitations are primarily manifested in two principal domains:

Primarily, the intrinsic dynamism and perpetual evolution of social media platforms, alongside the resultant information ecosystems, present a formidable challenge to encapsulating a stable snapshot of the infodemic landscape. The empirical research methodology deployed herein, while advantageous in illuminating phenomena and relationships at specific temporal junctures, falls short in tracing and comprehending the longitudinal trends and underlying mechanisms of information sharing intentions. To surmount this limitation, future studies might contemplate employing quantitative analytical methods such as mathematical modeling and system dynamics simulations. Such methodologies could facilitate the simulation and prediction of users' information sharing intentions and behaviors across time, thereby offering a deeper understanding of the pathways and patterns influenced by various factors.

Secondarily, in scrutinizing information sharing behaviors within the social media milieu, this study espoused a relatively macroscopic perspective, viewing social media as a homogenous entity. However, this approach neglected to sufficiently account for the distinctive characteristics of differing social media platforms. Indeed, various social media platforms—each endowed with its own unique user demographics, interaction patterns, and content types—may substantially impact users' information sharing behaviors. Henceforth, future research endeavors should aspire to a more granular examination of the specific features of diverse social media platforms and investigate how these features shape users' preferences and patterns of information sharing.

## Supporting information

**S1 Dataset.**
(PDF)

**S1 Appendix.**
(PDF)

## Author Contributions

**Conceptualization:** Rui Shi.

**Data curation:** Xiaoran Jia.

**Formal analysis:** Xiaoran Jia.

**Investigation:** Xiaoran Jia.

**Software:** Xiaoran Jia.

**Supervision:** Xiaoran Jia, Hao Wang.

**Validation:** Xiaoran Jia, Yuhan Hu.

**Visualization:** Xiaoran Jia.

**Writing – original draft:** Xiaoran Jia.

**Writing – review & editing:** Rui Shi.

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
