## [Decision Letter · Decision Letter 0]

29 Apr 2024

PONE-D-24-10651The media risk of infodemic in public health emergencies :Consequences and Mitigation ApproachesPLOS ONE

Dear Dr. Wang,

Thank you for submitting your manuscript to PLOS ONE. After careful consideration, we feel that it has merit but does not fully meet PLOS ONE’s publication criteria as it currently stands. Therefore, we invite you to submit a revised version of the manuscript that addresses the points raised during the review process.

We look forward to receiving your revised manuscript.

Kind regards,

Anat Gesser-Edelsburg, Ph.D.

Academic Editor

PLOS ONE

Journal Requirements:

   "Ministry of Education (20YJC860027), Natural Science Foundation of Hebei Province Youth Science Fund (G2021203014), and Funded by Science Research Project of Hebei Education Department (SQ2023105)"

Reviewers' comments:

Reviewer's Responses to Questions

**Comments to the Author**

1. Is the manuscript technically sound, and do the data support the conclusions?

Reviewer #1: Yes

Reviewer #2: Partly

2. Has the statistical analysis been performed appropriately and rigorously? 

Reviewer #1: Yes

Reviewer #2: Yes

3. Have the authors made all data underlying the findings in their manuscript fully available?

Reviewer #1: Yes

Reviewer #2: Yes

4. Is the manuscript presented in an intelligible fashion and written in standard English?

Reviewer #1: Yes

Reviewer #2: No

5. Review Comments to the Author

Reviewer #1: This manuscript covers an important issue to be combated for promoting social welfare: the misinformation spread in social networks. The issue becomes even more important in the public health context, as the study was developed around the COVID-19 pandemic and related infodemic.

The authors developed a structural model combining six hypotheses related to how individuals’ information and attention are drawn regarding health emergency risk perceptions and information sharing, how opinion leaders associate with individuals’ willingness for information sharing, individuals’ perception about the risks of information sharing and individuals’ perception about and the usefulness of information about health emergencies as perceived by individuals, playing a moderating role in attention to information and individual risks.

The authors detail the construction of these hypotheses based on the literature presented in their literature review section, so, this aspect of their research is strictly formalized, in my opinion, which is a desired characteristic.

Some elements to be clarified in the text:

1. About the questionnaire application, how the sampling process was performed? How did the authors reach the number of 395 questionnaires to be sent? What was the population? I recommend the authors explain in more detail the data collection process, providing information related to these questions.

2. The authors stated that “The path analysis was carried out using AMOS 24.0 software …”. The software is an IBM product dedicated to Structural Equations modeling and analysis, so I recommend the authors provide in the Method section an introduction to IBM Amos 24.0 software, and the pipeline of the analytical process used. Note that the Data Analysis section describes this process in terms of results, so, the task here is to formulate, for instance, a flowchart of the whole analytical process, discriminating each step, in terms of methods used.

3. Results and discussion are described in a “plain text” with no separation of subsections. I recommend the authors dedicate sections about theoretical and practical implications as well as the social consequences of their research. About the social consequences, I believe the paragraph started in line 402 of the manuscript (“Therefore, social media play an important role in the dissemination of information on public health emergencies. The adoption of suitable steps to avoid and contain the future deterioration …”) can be placed or be used as the base to create the new section.

4. In Conclusion, it is important to clarify what were the limitations, difficulties, and challenges faced by the authors during the development of their study. These elements are relevant to ensure replicability. It is also important to point out some directions for new studies continuing or based on the study the authors developed.

Reviewer #2: This paper tackles a relevant and timely topic, but I had a hard time reading and understanding what's going on.

-> Writing style and structure needs to be greatly improved for conveying the findings and takeaways of the work and before this work can be reviewed properly for its contributions.

-> Paper suffers from redundancy, with information repeated across different sections, making the paper much lengthier than it should be. Repetition also causes to dilute the key points of the work.

-> The literature review should be streamlined by focusing on the most relevant studies and avoiding excessive detail and definitions of the underlying concepts.

-> The flow of the paper needs major improvements. Sections and text feel disjointed, and the transitions are not clear. I struggled with making a connection between risk perception and information sharing behavior which are the key ideas of the paper.

-> The language seems to be convoluted and unduly complicated. It feels like I am reading translated text, as sentences seem to break midway followed by another idea or sentence. Additionally at some places, the sentences are too long don't seem to be conveying anything important.

-> The conclusion should be strengthened by emphasizing the unique contributions of the study, highlighting how the findings extend or augment the previous research and offer new perspectives on managing infodemics. The recommendations seem textbook version of existing suggestions for mitigating misinformation. The paper should include succinct but in-depth discussion of how these recommendations can be implemented effectively.

-> The paper's central theme seems to drift towards related but tangential topics, such as the general impact of social media on information dissemination, please try to maintain a sharper focus on the core theme.

-> For data collection, how were the target poulation selected. Moreover, most of the survey takers are highly educated which obviously creates a bias in the analysis.

-> Please try including informative figures to present key findings and helpwith understanding and engagement of the reader. The included figure on moderated mediation was confusing as well.

That said, it is good that authors seem to have spent extra effort in ensuring proper statistical treatment. For future submissions, please also outcome of regular regression models. The response data collected has been made avilable which is nice.

6. PLOS authors have the option to publish the peer review history of their article (what does this mean?). If published, this will include your full peer review and any attached files.

Reviewer #1: No

Reviewer #2: No

---

## [Author Response · Author response to Decision Letter 0]

31 May 2024

Response to the Reviewer 1’s comments

1.About the questionnaire application, how the sampling process was performed? How did the authors reach the number of 395 questionnaires to be sent? What was the population? I recommend the authors explain in more detail the data collection process, providing information related to these questions.

Author’s Response: Thank you for your astute observation!

We have incorporated an exhaustive exposition on the sampling survey methodology within our paper. To guarantee representative sampling, this study utilized a hybrid snowball and random sampling strategy for questionnaire dissemination.The process is specifically divided into three steps. To guarantee representative sampling, this study utilized a hybrid snowball and random sampling strategy for questionnaire dissemination. The aim was to achieve demographic balance across gender, age, education, and internet usage.The detailed methodology is outlined as follows:surveys were distributed via stratified random sampling among various age and gender groups in the researchers' social networks, asking participants to haphazardly pass the questionnaire to their friends to fill out. Data collection from demographic categories that were properly represented was then stopped once a preset amount of responses were received. In order to reduce the possibility of sample biases, extra stratified random sampling was carefully carried out among underrepresented groups.Lastly, to counterbalance the limitations inherent in snowball sampling, supplementary random sampling was conducted in university libraries, enterprises, institutions, and outdoor parks. Respondents were selectively approached to enhance diversity, and questionnaires were thoroughly explained to ensure comprehension, thereby improving response quality and validity.This multi-faceted approach was instrumental in compiling a well-balanced and informative dataset for the study.

Furthermore, the questionnaire was released online and distributed through WeChat and Credamo sample pools. Credamo is an integrated smart research platform specializing in questionnaire design, sample provision, and statistical analysis. It streamlines data collection and analysis processes for academic and corporate users. Offering end-to-end services from questionnaire creation, online deployment, data gathering to statistical modeling, Credamo boasts a million-level online participant pool, ensuring data authenticity and diversity. It extensively supports universities, enterprises, and research institutions, facilitating efficient research project execution.(please see lines 236-249). 

2.The authors stated that “The path analysis was carried out using AMOS 24.0 software …”. The software is an IBM product dedicated to Structural Equations modeling and analysis, so I recommend the authors provide in the Method section an introduction to IBM Amos 24.0 software, and the pipeline of the analytical process used. Note that the Data Analysis section describes this process in terms of results, so, the task here is to formulate, for instance, a flowchart of the whole analytical process, discriminating each step, in terms of methods used.

Author’s Response: Thank you for your suggestions! 

We have augmented the "Methods" section with a detailed explanation of the structural equation modeling approach and the specific process of model construction. (please see lines 192-199).In consideration of the logical structure of this paper, we have also included in the path analysis section a description of the AMOS 24.0 software usage, along with how this tool is employed to validate the path coefficients of the structural equation models,highlighting its capabilities in Structural Equation Modeling (SEM) and its relevance to our research. (please see lines 284-297).This paper endeavors to employ the AMOS 24.0 software, a prominent tool in SEM analysis, to calibrate and refine a theoretical model explicating the determinants influencing the information sharing willingness on public health emergencies.

3.Results and discussion are described in a “plain text” with no separation of subsections. I recommend the authors dedicate sections about theoretical and practical implications as well as the social consequences of their research. About the social consequences, I believe the paragraph started in line 402 of the manuscript (“Therefore, social media play an important role in the dissemination of information on public health emergencies. The adoption of suitable steps to avoid and contain the future deterioration …”) can be placed or be used as the base to create the new section.

Author’s Response: Thank you for raising these important points!

We wholeheartedly agree with your suggestion to dedicate separate sections for discussing the theoretical and practical implications of our research, as well as its societal consequences. We have restructured the "Discussion" section, segregating it into distinct segments: one concentrating on the theoretical implications, another emphasizing practical applications, and a newly-added chapter devoted exclusively to examining the societal implications of our research findings.(please see lines 403-435).

Furthermore, we have relocated the paragraph started in line 402 of the original manuscript (“Therefore, social media play an important role in the dissemination of information on public health emergencies. The adoption of suitable steps to avoid and contain the future deterioration …”)to the Recommendations chapter, where we have built upon it to propose more targeted measures.(please see lines 437-464).

We believe this reorganization will significantly strengthen the narrative flow and emphasize the comprehensive significance of our work. We appreciate your guidance in this enhancement process and look forward to implementing these changes to elevate our manuscript's overall quality.

4.In Conclusion, it is important to clarify what were the limitations, difficulties, and challenges faced by the authors during the development of their study. These elements are relevant to ensure replicability. It is also important to point out some directions for new studies continuing or based on the study the authors developed.

Author’s Response: Thank you for your constructive feedback. 

Thank you for your thoughtful remarks which have greatly contributed to enhancing the rigor and utility of our study. We fully concur with your perspective on the importance of acknowledging limitations, difficulties, and challenges encountered during our research, as these aspects are crucial for promoting methodological transparency and facilitating future replication attempts.

In light of your feedback, we intend to enrich our conclusion section by explicitly addressing the following points:First, the challenge lies in capturing data amidst the dynamism of social media, where the rapidly evolving landscape of information epidemics and the sheer volume of data hinder precise assessments of long-term impacts. Second, the complexity in measuring risk perception and information-sharing behaviors arises due to the diversity in individual psychology, subjective interpretations of risk, and inconsistencies stemming from varied cultural backgrounds and levels of media literacy, all of which limit the generalizability of the study findings.(please see lines 466-486).

Response to the Reviewer 2’s comments

1.Writing style and structure needs to be greatly improved for conveying the findings and takeaways of the work and before this work can be reviewed properly for its contributions.

Author’s Response: Thank you for your suggestions! 

Regarding your suggestions on the style and structure of my paper, we have given them thoughtful consideration and carried out the necessary amendments. Throughout the revision process, we meticulously examined your review comments and also drew insights from dozens of articles published in the journal, striving to adjust the style and structure while preserving the integrity and accuracy of the content.

In terms of style, we have placed greater emphasis on clarity and logical coherence, striving to make the narrative of the paper more fluid and natural, thereby enhancing its comprehensibility. Additionally, we have reviewed and revised the technical terminology and expressions used in the paper to ensure their precision and consistency. For example, in the Introduction section, we have built upon the integration of the research background and recent studies to explicitly state the objectives and anticipated outcomes of our research.(please see lines 12-80). Within the section on Hypotheses develpoment, we have consciously refrained from delving too deeply into the theory of risk perception, instead redirecting the focus towards a meticulous presentation of each hypothesis. This shift ensures that the prime objectives and forecasted results of the study are conveyed with clarity, thereby enhancing readers' grasp of the central questions and the crux of our investigation. By doing so, a solid foundation is established for the following analytical examinations and discussions.(please see lines 81-189).

As for the structure, we realigned the sections and rearranged paragraphs based on your guidance, with the aim of better illustrating the logical relationships and argumentative progression within the paper. Efforts were also made to eliminate redundancy and repetitive content, rendering the paper more concise and compact. In the Discussion section, we uncovered a significant positive correlation between opinion leaders, information attention, and the willingness to share information, endorsing our hypotheses and highlighting the substantial influence of social media on information sharing. Furthermore, the incorporation of risk perception illuminated the intricacies of these relationships, demonstrating its mediating role in the link between opinion leaders, information attention, and the willingness to share. Lastly, the moderating effect of perceived usefulness on the relationship between information attention and sharing willingness was validated, further enriching our findings.(please see lines 327-401). In the Conclusions, this study theoretically enriches the understanding of the relationship between social media, risk perception, and the willingness to share information, offering a fresh perspective on their underlying mechanisms. Practically, our findings can serve as a basis for policymakers, enabling them to better comprehend and address pertinent issues.(please see lines 402-435). In the Recommendations section, we have put forth more specific measures, such as: enhance transparency and credibility, risk communication strategies, empower opinion leaders and influencers and research and monitoring.(please see lines 436-464). 

2.Paper suffers from redundancy, with information repeated across different sections, making the paper much lengthier than it should be. Repetition also causes to dilute the key points of the work.

Author’s Response: Thank you for your meticulous review and for pointing out the issue of redundancy in our paper.

We fully understand that unnecessary repetition can undermine the clarity and impact of our key findings. The original manuscript's literature review excessively emphasized risk perception, resulting in repetitive content. Additionally, the section overlappingly discussed the research hypotheses both in the literature review and conclusion, unduly extending the paper's length and clouding its primary focus. To tackle these issues, comprehensive revisions were executed as follows:

Firstly, we integrated repeated information originally included for reinforcement, eliminating superfluous explanations of concepts and methodologies to promote clarity and brevity.

Secondly, the introduction was enriched to encompass not only the research backdrop but also a synopsis of the latest developments within the discipline, facilitating readers' comprehension of the field's ongoing progress and existing gaps. This sets a pertinent context for the study.(please see lines 12-80).

Next, the conclusion was reoriented to concentrate specifically on how social media influences users' risk perception and, subsequently, their inclination to share information, staying aligned with the article's core subject matter. By pruning content that did not advance the central thesis directly, we intensified the focus on our pivotal arguments and discoveries.(please see lines 403-435).

Lastly, to uphold coherence and prevent redundancy when similar points arose in different contexts, these sections were either restated in fresh terms or cross-referenced to prior explanations, maintaining continuity without duplicating text verbatim.

3.The literature review should be streamlined by focusing on the most relevant studies and avoiding excessive detail and definitions of the underlying concepts.

Author’s Response: Thank you for your suggestions! 

After meticulously reviewing your reviewer comments, we recognized the need to enhance the logical structure of the article. Consequently, we have incorporated discussions on the most recent research in the introduction, with the aim of enhancing the narrative's coherence and readability. Simultaneously, we have summarized and synthesized relevant literature, steering clear of unnecessary elaboration on fundamental concepts. Furthermore, in integrating these latest findings, we have delved into authoritative publications from recent times, meticulously analyzing the principal arguments put forth in these works.(please see Introduction 2-80).My objective was to integrate these latest findings closely with my research topic, thereby presenting a more comprehensive and precise depiction of the current dynamics and cutting-edge trends in the field. We are confident that the inclusion of these recent studies will facilitate readers' understanding of the latest developments in this domain and augment the academic worth and impact of the paper.

4.The flow of the paper needs major improvements. Sections and text feel disjointed, and the transitions are not clear. I struggled with making a connection between risk perception and information sharing behavior which are the key ideas of the paper.

Author’s Response: Thank you for your insightful feedback. 

Having carefully read through your review report, we gained a profound understanding of the areas in which my paper required improvement regarding structure and argumentation.In response to your comments on the structure and organization of chapters, we have conducted thorough revisions. This round of modifications particularly focused on adjusting the chapter layout to better adhere to logic and academic standards. 

Since structural equation modeling (SEM) typically relies on a theoretical framework or assumptions, in the section Hypotheses Development, we meticulously analyzed each hypothesis's independent and dependent variables, along with their direct or indirect effects, thereby enhancing the interpretability of our findings.(please see lines 81-189).

Within the Research Method and Design section, we first introduced the concept of SEM and presented the theoretical model guiding our study.(please see lines 191-205).To measure latent variables that cannot be directly observed, we employed a questionnaire survey method, transforming these latent variables into quantifiable observed data, and meticulously detailed the questionnaire design process.(please see lines 211-234). We then outlined the questionnaire distribution procedure, including pilot testing, the method of distribution, and the sampling process for the survey.(please see lines 235-262).

In the Data Analysis section, we initiated with reporting the results of reliability and validity tests.(please see lines 264-282). Following this, path analysis and hypothesis testing were conducted based on the established foundation.(please see lines 283-297). Lastly, we examined the mediating effect of risk perception and the moderating role of perceived usefulness.(please see lines 299-325).

The Discussion section thoroughly dissects the outcomes of each path analysis, transparently demonstrating whether the data supports these relationships, thereby validating our 

---

## [Decision Letter · Decision Letter 1]

2 Jul 2024

PONE-D-24-10651R1The media risk of infodemic in public health emergencies :Consequences and Mitigation ApproachesPLOS ONE

Dear Dr. Wang,

Thank you for submitting your manuscript to PLOS ONE. After careful consideration, we feel that it has merit but does not fully meet PLOS ONE’s publication criteria as it currently stands. Therefore, we invite you to submit a revised version of the manuscript that addresses the points raised during the review process.

We look forward to receiving your revised manuscript.

Kind regards,

Prof. Anat Gesser-Edelsburg, Ph.D.

Academic Editor

PLOS ONE

Reviewers' comments:

Reviewer's Responses to Questions

**Comments to the Author**

1. If the authors have adequately addressed your comments raised in a previous round of review and you feel that this manuscript is now acceptable for publication, you may indicate that here to bypass the “Comments to the Author” section, enter your conflict of interest statement in the “Confidential to Editor” section, and submit your "Accept" recommendation.

Reviewer #1: All comments have been addressed

Reviewer #3: (No Response)

2. Is the manuscript technically sound, and do the data support the conclusions?

Reviewer #1: Yes

Reviewer #3: Yes

3. Has the statistical analysis been performed appropriately and rigorously? 

Reviewer #1: Yes

Reviewer #3: Yes

4. Have the authors made all data underlying the findings in their manuscript fully available?

Reviewer #1: Yes

Reviewer #3: Yes

5. Is the manuscript presented in an intelligible fashion and written in standard English?

Reviewer #1: Yes

Reviewer #3: No

6. Review Comments to the Author

Reviewer #1: The authors presented details of what they applied to the manuscript to meet the recommendations made in the previous round. There were significant changes to the text, demonstrating that they dedicated considerable time and effort to improving their material:

1. Details about the questionnaire construction and sample selection process were provided at the beginning of the Sample and Data Collection section.

2. The authors also improved their text on SEM, before introducing Figure 1, which demonstrates the structural model used with the constructs involved. A more detailed presentation was also made about the software used and the process involved in hypothesis testing.

3. In the Conclusions, three paragraphs were developed dedicated to commenting on the theoretical and practical implications of the work, as well as the social impact involved.

4. In general, the Conclusions were modified, and I believe that the impact of these modifications was quite positive, separating the authors' recommendations as well as presenting the limitations of the research.

Within the social repercussions, I would like to read about the authors' opinion on the psychological effects of misinformation on the population concerning increased anxiety, especially considering critical situations (such as what happened during the COVID-19 pandemic).

• How does the analysis of the infodemic scenario contribute to assisting public health authorities in developing strategies capable of mitigating the stress effects of these situations? What technological tools can be used?

• For example: how can psychometrics be used in these situations?

Finally, I recommend that the authors conduct a general review of the use of the English language. I found, for example, the use of the compact form “it’s” when it is formally recommended to use “it is” (see line 442). There is a single occurrence of this use, from what I found through a search throughout the text, however, I strongly recommend a general revision to leave the text in a "standard" format for English usage.

Reviewer #3: The article explores the risk of infodemics in public health emergencies and their mitigation methods, which is a very important and highly relevant research topic, especially in the current context of frequent global epidemics. In detail, the article constructs a novel risk perception theoretical model and employs structural equation modeling for empirical analysis. This scientific and rational research design aids in a deeper understanding of social network users' information-sharing behavior during public health emergencies. The article provides a comprehensive review of relevant literature and, based on this, proposes reasonable research hypotheses, demonstrating the authors' deep understanding and mastery of the research field. Moreover, the article not only offers new insights in the field of risk perception and infodemics, making significant theoretical contributions, but also proposes specific policy recommendations and information governance strategies, providing strong practical guidance. No critical flaws in the research content can be pointed out, and it is well-documented. However, I have several points for consideration outlined below.

1. The English of this manuscript definitely needs to be polished or rechecked at least. For example, in lines 111 and 307, "Crown pneumonia epidemic" and "new crown outbreak" seem to refer to COVID-19, but these expressions are completely incorrect in English.

2. Please describe the process of distributing the questionnaires in more detail, including the characteristic of the respondents. For example, were the questionnaires distributed online or offline? Was the distribution process random? Did the respondents belong to a specific group or organization? Explain any potential biases in the data collection process and how these biases were controlled.

3. Provide more detail on the demographic characteristics of the sample and justify the sample size.

4. The methodology section lacks details on the questionnaire development process. How were the items selected or adapted from existing scales? Were any pilot studies conducted to assess the validity and reliability of the questionnaire?

5. Using the words “Cronbach's alpha coefficient” or “Cronbach's α” instead of “Cronbach's”.

6. The authors briefly introduce the aim of the study in the abstract. However, I suggest to provide a more detailed study aim in the introduction section.

7. Discuss potential limitations of the study in more detail, including any methodological constraints and generalizability issues.

8. In “Results of reliability and validity” section, introduce what software was utilized for data analyzing.

7. PLOS authors have the option to publish the peer review history of their article (what does this mean?). If published, this will include your full peer review and any attached files.

Reviewer #1: No

Reviewer #3: **Yes: **Mingxin Liu

---

## [Author Response · Author response to Decision Letter 1]

7 Jul 2024

Response to the Reviewer 1’s comments

Dear Reviewer,

We sincerely express our gratitude for the reviewer's demonstration of professional rigor and a high sense of responsibility towards academic pursuits. Your endorsement of our preliminary revision efforts is greatly honored. In response to your subsequent insightful comments and observations, we have meticulously revised and fine-tuned the manuscript, ensuring that all concerns have been satisfactorily addressed.

Please review the attached detailed responses to your concerns, along with the revisions we've carefully made to address each point you've raised.

1.How does the analysis of the infodemic scenario contribute to assisting public health authorities in developing strategies capable of mitigating the stress effects of these situations? What technological tools can be used?For example: how can psychometrics be used in these situations?

Author’s Response: Thank you for your astute observation!

In response to the suggestion to incorporate a discussion on the utilization of psychometrics to assist public health authorities in devising strategies to mitigate the stress effects of misinformation, we have revised the manuscript accordingly: Social media play an important role in the dissemination of information on public health emergencies. The implementation of appropriate measures to prevent and mitigate the escalation of public health crises is significantly enhanced by access to accurate and factual information. This enables individuals to develop a realistic understanding of the risks involved and gauge the seriousness of such emergencies accurately. However, the proliferation of false information can lead to heightened fear and panic, causing individuals to misinterpret the severity of the situation and adopt harmful behaviors based on misinformation. This not only undermines public health efforts but also takes a toll on mental health. By understanding the dynamics of information spread, public health authorities can develop targeted interventions. (please see lines 448-457). 

Psychometric analysis has been incorporated into the specific measures. Utilizing psychometrics enables the identification of vulnerable populations who may be more susceptible to misinformation and its associated stress effects. By understanding the psychological profiles of individuals prone to anxiety and misinformation acceptance, targeted interventions can be designed. These might include mental health support services, fact-checking resources, and community-led initiatives that foster critical thinking and resilience. (please see lines 471-476). The revised section highlights the importance of psychometric analysis in understanding the cognitive and emotional responses of individuals to misinformation. 

2.Finally, I recommend that the authors conduct a general review of the use of the English language. I found, for example, the use of the compact form “it’s” when it is formally recommended to use “it is” (see line 442). There is a single occurrence of this use, from what I found through a search throughout the text, however, I strongly recommend a general revision to leave the text in a "standard" format for English usage.

Author’s Response: We would like to express our sincere gratitude for the reviewer's meticulous attention to detail and constructive feedback regarding the usage of the English language in our manuscript.

In accordance with your insightful recommendation, we have embarked on a comprehensive examination of the entire manuscript to ensure conformity with standard language usage. Particularly, concerning the issue you pinpointed at line 442 involving the casual form "it's", we have taken corrective action. 

Moreover, taking the reviewer's advice to heart, we have decided to conduct a thorough review of the entire text to ensure that all instances conform to standard English usage. For instance, in the Introduction section, the original expression was "Gunther Eysenbach first proposed the idea of infodemiology in 2002 with the goal of examining the distribution and determinants of correct and incorrect information related to public health, directing patients and experts in the area toward reliable, accurate information", which has been revised to "Gunther Eysenbach initally proposed the term of infodemiology in 2002, aiming to investigate the distribution and determinants of health-related information, both accurate and erroneous, guiding patients and professionals in the field toward trustworthy and precise information". (please see lines15-19). We have also revised the expressions in the Introduction and Hypotheses development. This will include a careful examination of grammar, syntax, and style to eliminate any casual language or colloquialisms that may inadvertently deviate from formal academic writing conventions.

We appreciate the reviewer's guidance in helping us improve the quality of our submission. We look forward to incorporating these changes and resubmitting a polished and refined version of our paper for your consideration.

Response to the Reviewer 3’s comments

Dear Reviewer,

We are immensely grateful for the time and expertise you are dedicating to reviewing our manuscript. We are aware that you might be evaluating the original version of our manuscript. It's important to note that a revised version has already been developed, thoroughly addressing the concerns raised by previous reviewers. Building upon this improved draft, we have now incorporated further amendments in direct response to your suggestions, ensuring that our manuscript is finely tuned to meet the high standards of scholarly discourse.

Enclosed below are our detailed responses to your concerns, alongside the modifications we've made in response to your valuable feedback.

1.The English of this manuscript definitely needs to be polished or rechecked at least. For example, in lines 111 and 307, "Crown pneumonia epidemic" and "new crown outbreak" seem to refer to COVID-19, but these expressions are completely incorrect in English.

Author’s Response: Thank you for your careful reading of our manuscript and for pointing out areas where the English language usage can be improved. We appreciate your attention to detail and acknowledge that the expressions "Crown pneumonia epidemic" and "new crown outbreak," as noted in lines 111 and 307, were indeed imprecise and non-standard in English.

In response to your feedback, we have taken the following actions:

1.We have corrected the inaccurate phrases to "COVID-19" respectively, ensuring that our terminology aligns with the accepted English nomenclature.

2.Beyond addressing the specific issues you highlighted, we have also engaged a native English-speaking professional editor to review and polish the entire manuscript. This thorough revision aims to enhance the clarity, coherence, and overall standard of English throughout the text.

2.Please describe the process of distributing the questionnaires in more detail, including the characteristic of the respondents. For example, were the questionnaires distributed online or offline? Was the distribution process random? Did the respondents belong to a specific group or organization? Explain any potential biases in the data collection process and how these biases were controlled.

Author’s Response: Thank you for your insightful feedback. 

Between March 1 and March 7, 2024, the questionnaires were released online through WeChat and the Credamo(https://www.credamo.com/home.html) platform, aiming to maximize accessibility and inclusivity by reaching out to a diverse participant base digitally. (please see lines 241-243).

To guarantee the representativeness of the research sample, this study adopted an integrated sampling strategy that artfully combined snowball sampling with random sampling methods. This approach aimed to construct a sample framework that accurately reflects the structure of internet users, particularly in terms of gender, age, and educational background. The detailed methodology is outlined as follows: surveys were distributed via stratified random sampling among various age and gender groups in the researchers' social networks, asking participants to haphazardly pass the questionnaire to their friends to fill out. Data collection from demographic categories that were properly represented was then stopped once a preset amount of responses were received. Lastly, to counteract the bias of snowball sampling, targeted random distribution of questionnaires was conducted on the Credamo platform for underrepresented groups. Respondent selection enhanced sample diversity, while accompanying detailed questionnaire explanations secured participant understanding, boosting response quality and data validity, and solidifying research reliability. This multi-faceted approach was instrumental in compiling a well-balanced and informative dataset for the study.(please see lines 244-259).

The respondents were not confined to any specific group, the age range and educational backgrounds of the participants were notably wide, encompassing individuals from various life stages and academic achievements, which showcase the breadth and diversity of the survey population. (please see lines 264-273).

Potential biases in the data collection process were carefully considered and addressed. For instance, to mitigate response bias, we ensured the anonymity and confidentiality of the participants, encouraging honest and unbiased responses. Additionally, we took measures to reduce sampling bias by reaching out to underrepresented groups and employing snowball sampling techniques when necessary. (please see lines 253-268).

3.Provide more detail on the demographic characteristics of the sample and justify the sample size.

Author’s Response: We appreciate your suggestion to provide more detail on the demographic characteristics of our sample and to further justify the sample size.

In response to your comment, we have revised the "Sample and data collection" section of the manuscript to include a more comprehensive breakdown of the demographic variables of our sample. Specifically, we have added detailed information about the gender, age distribution, and educational background of the participants. This additional information will help readers better understand the composition of our sample and its relevance to the research questions. (please see lines 264-273 and Table 1.).

Regarding the sample size justification, for exploring variable relationships via structural equation modeling, a minimum sample size of 200 is advisable, ideally exceeding ten times the number of observed variables for robust analyses [1]. In this study, 395 questionnaires in total were sent; 346 of those were deemed legitimate, and the effective recovery rate was 87.59%. (please see lines 260-263).

References:

[1] Bentler P M, Chou C P. Practical issues in structural modeling[J]. Sociological methods & research, 1987, 16(1): 78-117.

4.The methodology section lacks details on the questionnaire development process. How were the items selected or adapted from existing scales? Were any pilot studies conducted to assess the validity and reliability of the questionnaire?

Author’s Response: We sincerely appreciate the reviewer's thorough evaluation and insightful comments regarding the methodology section of our manuscript.

To address the reviewer's concerns, we have revised the manuscript to include a detailed description of the questionnaire development process. (please see lines 206-233).

We drew upon established scales (please see lines 206-212) and adapted certain items to align with the specific objectives of our study, ensuring that the questionnaire captured the multifaceted nature of information dissemination in the context of health crises. In concordance with the study hypotheses and mindful of the exigencies dictated by the COVID-19, a meticulous revision and contextualization of survey items sourced from antecedent investigations was undertaken. Precisely, within the rubric measuring information attention, pandemic-specific nuances were integrated, exemplified by the item reflecting vigilance over the evolving dynamics pertinent to the COVID-19. Concerning risk perception, the awareness of potential perils intrinsic to the COVID-19 was emphasized through the calibration of relevant questions. Furthermore, in the evaluation of proclivity towards information sharing willingness, the pertinence of shared information to the COVID-19 milieu was explicitly delineated, as embodied in the item highlighting the propensity to disseminate or alert others upon receipt of updated the COVID-19 information. Through these targeted refinements, the questionnaire's contemporaneity and situational specificity were assured, thereby facilitating a more accurate delineation of participant behavioral paradigms and psychological orientations situated within the COVID-19 framework. (please see lines 212-225).

To assess the validity and reliability of the questionnaire, we conducted a pilot study involving a sample of 104 participants who were demographically similar to our main study population. The pilot study enabled us to evaluate the clarity, comprehensibility, and relevance of the items, as well as to test the internal consistency and construct validity of the questionnaire. Based on the feedback received and statistical analyses, minor adjustments were made to refine the wording of some items and to ensure that the questionnaire was both valid and reliable for measuring the constructs of interest. Furthermore, we utilized Cronbach's alpha to measure the internal consistency of the scale, which yielded satisfactory results above the conventional threshold of 0.7, indicating that the items within each subscale were sufficiently correlated to measure the intended constructs reliably. (please see lines 234-239).

5.Using the words “Cronbach's alpha coefficient” or “Cronbach's α” instead of “Cronbach's”.

Author’s Response: We are grateful for the reviewer's meticulous attention to detail in our manuscript. 

We have since reviewed the entire manuscript and have rectified this inconsistency. Now, wherever applicable, we have used the "Cronbach's α" to refer to the measure of internal consistency. (please see lines 281).This change ensures that our terminology is scientifically accurate and consistent with the conventions of psychological and statistical research.

6.The authors briefly introduce the aim of the study in the abstract. However, I suggest to provide a more detailed study aim in the introduction section.

Author’s Response: We appreciate the reviewer's thoughtful feedback regarding the clarity and depth of our study's aims as presented in the abstract and introduction sections.We recognize the importance of clearly articulating the research objectives to provide a solid foundation for the study and guide readers' expectations. 

Following the reviewer's suggestion, we have revised the abstract to offer a more detailed and explicit statement of the study's aims. The revised abstract now highlights the specific gaps in knowledge that our research seeks to address and outlines the primary research questions and hypotheses we intend to explore. This enhanced clarity will help readers immediately grasp the purpose and significance of our study. (please see lines 66-78).

Additionally, we have expanded the introduction section to provide a comprehensive rationale for our research. We have included a detailed discussion of the background, including a review of the existing literature, which underscores the need for our investigation. We have also delineated the specific objectives of our study, explaining how they contribute to the broader field and what new insights we anticipate our research will provide. (please see lines 2-65).

7.Discuss potential limitations of the study in more detail, including any methodological constraints and generalizability issues.

Author’s Response: We are grateful for the reviewer's constructive suggestions regarding the discussion of our study's potential limitations.

In response to the reviewer's suggestion, we have expanded the "Research limitations" section of our manuscript. We have now provided a more detailed and nuanced discussion of the 

---

## [Decision Letter · Decision Letter 2]

17 Jul 2024

The media risk of infodemic in public health emergencies :Consequences and Mitigation Approaches

PONE-D-24-10651R2

Dear Dr. Wang,

We’re pleased to inform you that your manuscript has been judged scientifically suitable for publication and will be formally accepted for publication once it meets all outstanding technical requirements.

Kind regards,

Prof. Anat Gesser-Edelsburg, Ph.D.

Academic Editor

PLOS ONE

Additional Editor Comments (optional):

Reviewers' comments:

Reviewer's Responses to Questions

**Comments to the Author**

1. If the authors have adequately addressed your comments raised in a previous round of review and you feel that this manuscript is now acceptable for publication, you may indicate that here to bypass the “Comments to the Author” section, enter your conflict of interest statement in the “Confidential to Editor” section, and submit your "Accept" recommendation.

Reviewer #1: All comments have been addressed

Reviewer #3: All comments have been addressed

2. Is the manuscript technically sound, and do the data support the conclusions?

Reviewer #1: Yes

Reviewer #3: Yes

3. Has the statistical analysis been performed appropriately and rigorously? 

Reviewer #1: Yes

Reviewer #3: Yes

4. Have the authors made all data underlying the findings in their manuscript fully available?

Reviewer #1: Yes

Reviewer #3: Yes

5. Is the manuscript presented in an intelligible fashion and written in standard English?

Reviewer #1: Yes

Reviewer #3: Yes

6. Review Comments to the Author

Reviewer #1: In this new round, the authors addressed the final questions I had raised.

I therefore understand that the work is adequate in accordance with the indications I made in previous reviews.

Reviewer #3: I believe the authors have addressed my comments and those of Reviewer 1 very well. Therefore, I consider this manuscript to meet the publication requirements of PLOS ONE.

7. PLOS authors have the option to publish the peer review history of their article (what does this mean?). If published, this will include your full peer review and any attached files.

Reviewer #1: No

Reviewer #3: **Yes: **Mingxin Liu

---

## [Editor Report · Acceptance letter]

2 Sep 2024

PONE-D-24-10651R2 

PLOS ONE

Dear Dr. Wang, 

I'm pleased to inform you that your manuscript has been deemed suitable for publication in PLOS ONE. Congratulations! Your manuscript is now being handed over to our production team.

Kind regards, 

on behalf of

Prof. Anat Gesser-Edelsburg 

Academic Editor

PLOS ONE